# HARNESSING ON-DEVICE LARGE LANGUAGE MODEL: EMPIRICAL RESULTS AND IMPLICATIONS FOR AI PC

## ABSTRACT

The increasing deployment of Large Language Models (LLMs) on edge devices, driven by model advancements and hardware improvements, offers significant privacy benefits. However, these on-device LLMs inherently face performance limitations due to reduced model capacity and necessary compression techniques. To address this, we introduce a systematic methodology—encompassing model capability, development efficiency, and system resources—for evaluating on-device LLMs. Our comprehensive evaluation, encompassing models from 0.5B to 14B parameters and seven post-training quantization (PTQ) methods on commodity laptops, yields several critical insights: 1) System-level metrics exhibit near-linear scaling with effective bits-per-weight (BPW). 2) A practical threshold exists around ∼3.5 effective BPW, larger models subjected to low-bit quantization consistently outperform smaller models utilizing higher bit-precision. 3) As model size decreases, the primary performance bottleneck potentially shifts from computation to communication. 4) Determined by low-level implementation specifics power consumption on CPU, computation-intensive operations spend more power than memory-intensive ones. These insights offer practical guidelines for the efficient deployment and optimized configuration of LLMs on resource-constrained edge devices. Our codebase is available at `https://anonymous.4open.science/r/LLMOnDevice/`.

## 1 INTRODUCTION

Large language models (LLMs) have revolutionized modern applications through their advanced text-generation capabilities (Achiam et al., 2023; Liu et al., 2024). While traditionally cloud-dependent, a paradigm shift towards on-device deployment is emerging (Chen & Li, 2024; Hu et al., 2024), enabled by breakthroughs in efficient model design (Zhu et al., 2024) and advancements in edge computing hardware. This transition also addresses critical privacy concerns: on-device execution eliminates remote data transmission, positioning these models as essential tools for privacy-sensitive domains such as healthcare (Thirunavukarasu et al., 2023) and finance (Wu et al., 2023).

Despite these advantages, on-device LLMs face inherent limitations due to their reduced parameter count and the use of compression techniques like quantization and pruning (Zhu et al., 2024), which restrict their performance potential (Xu et al., 2024). Nonetheless, studies show they perform routine tasks—such as text summarization, intent recognition, and local query resolution—with adequate competence (Team et al., 2025). However, the lack of a comprehensive evaluation leaves their true capabilities underexplored. As these models evolve rapidly, establishing a comprehensive evaluation framework is crucial to ensure reliability, scalability, and alignment with user needs.

Numerous methods and readily available benchmarks have been proposed for evaluating the practical viability of LLMs for on-device deployment. For instance, MLPerf Client (MLCommons Association, 2025) evaluates Time to First Token (TTFT) and Tokens Per Second (TPS) for the 4-bit quantized Llama 2 model (Touvron et al., 2023) across tasks inspired by real-world applications, such as content generation and text summarization. Similarly, LocalScore (Pais, 2025) assesses pre-filling throughput, text generation throughput, and TTFT for 4-bit quantized models of 1B, 8B, and 14B parameters. Power consumption metrics have also been explored, as demonstrated by Stevens (2024) who measure consumption in Watt-hours.

However, a common limitation is that most existing works typically concentrate on a subset of metrics, often evaluating accuracy, efficiency, or power consumption in isolation. While Husom et al. (2025) concurrently evaluated output accuracy, inference performance, and energy efficiency, their analysis lacks a detailed examination of how these metrics are influenced by varying workloads (e.g., model size, token length) and specific methodologies (e.g., CPU operator implementations for quantization techniques).

To address these limitations, we propose a tripartite evaluation framework for on-device LLM inference that systematically considers: 1) *Model capability:* assessing task-specific accuracy using standardized benchmarks. 2) *Deployment efficiency:* quantifying generation throughput and latency under practical hardware constraints. 3) *System resource utilization:* analyzing resource consumption and potential contention, particularly in environments with concurrent applications. This multidimensional framework aims to provide a holistic understanding of the scenarios where, and methodologies by which, on-device LLMs can effectively serve as alternatives to cloud-based solutions.

Our investigation employs a representative consumer-grade Windows laptop with a single CPU and 16GB RAM, mirroring typical user hardware configurations. Through rigorous experimentation, we address three pivotal questions: 1) We establish the maximum model size that maintains acceptable system responsiveness during concurrent multitasking. 2) We dissect the trade-offs between quantization precision levels (4-bit to 8-bit) across our evaluation dimensions, revealing nonlinear relationships between bit-width reduction and performance degradation. 3) We compare different post-training quantization methods, demonstrating how algorithmic choices affect deployment outcomes. These findings provide actionable insights for optimizing the balance between model efficacy and resource efficiency in edge computing environments. Our primary contributions are:

**Comprehensive Evaluation of On-Device LLM Inference:** We conduct an extensive empirical study of LLM inference on edge devices using the Llama.cpp framework. The investigation encompasses eight LLMs of diverse parameter scales, subjected to seven distinct quantization methods. Model capability is benchmarked across five open-source datasets, while inference efficiency and resource utilization are assessed under four different input token lengths. This comprehensive methodology spans diverse model architectures, quantization levels, task types, and workloads.

**Deployment Insights for Practical Scenarios:** We offer actionable insights into the deployment of LLMs in production environments on edge devices. This includes an analysis of the trade-offs between task accuracy and deployment efficiency, guiding optimal model and quantization selection.

**Guidance for LLM Inference Architecture and Acceleration:** We provide recommendations for architectural optimizations pertinent to LLM inference frameworks. We offer insights for accelerating LLM inference on resource-constrained edge devices by analyzing potential bottlenecks, particularly the increasing communication overheads associated with model sizes.

## 2 PRELIMINARIES

**Quantization.** Deploying LLMs on resource-constrained devices requires model compression (Zhu et al., 2024), with post-training quantization being a leading strategy due to its ability to reduce model size and computational cost with minimal performance impact. A primary framework categorizes quantization methods into *symmetric* and *asymmetric* variants based on their alignment with the origin. Following ggml-org (2024), let $x \in \mathbb{R}$ be an original (pre-quantization) value and $q$ its quantized counterpart, using $n \in \mathbb{N}^+$ bits of precision, symmetric quantization employs $n$-bit signed integers, where the discrete quantized value $q$ lies in the range $\{-2^{n-1}, \ldots, 2^{n-1} - 1\}$. A scaling factor $s \in \mathbb{R}$ is computed as $s = \frac{|x|_{\max}}{2^{n-1}}$, where $|x|_{\max}$ is the maximum absolute value of $x$. Denoting rounding by Round$(\cdot)$, $q$ is computed as

$$q = \max\left(-2^{n-1}, \min\left(\text{Round}\left(\tfrac{x}{s}\right), 2^{n-1} - 1\right)\right).$$

Asymmetric quantization employs $n$-bit unsigned integers, resulting in $q \in \{0, \ldots, 2^n - 1\}$. Let $m = x_{\min}$ and $x_{\max}$ be the minimum and maximum values of the input data $x$, respectively. The scaling factor $s$, which maps the input range $[m, x_{\max}]$ to approximately $[0, 2^n - 1]$, is computed as $s = \frac{x_{\max} - m}{2^n - 1}$. The input $x$ is then quantized as $q = \max\left(0, \min\left(\text{Round}(\tfrac{x-m}{s}), 2^n - 1\right)\right)$.

**Quantization methods in `llama.cpp`.** llama.cpp (ggml-org, 2024) stands out as a crucial tool in the landscape of on-device LLM research and development due to its open-source and

cross-platform nature. The quantization schemes `n_0` and `n_k` implemented in `llama.cpp` is adopted in our study. The `n_0` method employs symmetric quantization, where weights are scaled uniformly using a zero-centered range, eliminating the need for zero-point offsets. In contrast, the `n_k` framework uses asymmetric strategy and further extends through five synergistic enhancements: 1) Hierarchical parameter grouping, which recursively quantizes scale $s$ and offset $m$ metadata to reduce overhead. 2) Activation-guided importance matrices that prioritize high-impact dimensions during quantization, mitigating accuracy loss from skewed weight distributions. 3) Post-quantization convex optimization to refine scales and zero-points by minimizing layer-wise reconstruction error. 4) Perturbative search algorithms that iteratively adjust quantized values to escape local minima, improving parameter recovery. 5) Heterogeneous bit allocation, assigning different precision to different weight matrices.

## 3 METHODOLOGY

This section outlines the evaluation methodology employed for on-device LLMs, encompassing the selection criteria for quantization methods and models, and the evaluation framework of 1) model capability, 2) deployment efficiency, and 3) system resource utilization, respectively.

### 3.1 MODEL AND QUANTIZATION SELECTION

This study considers the Qwen 2.5 (Team, 2024b) and Llama 3 (Team, 2024a) series of LLMs due to their widespread adoption and popularity within the research community and industry. Specifically, Qwen 2.5 models with 0.5B, 1.5B, 3B, 7B, 14B parameters and Llama 3 models with 1.5B, 3B, 8B parameters are selected for evaluation. The upper limit of 14B parameters was chosen as it represents the largest model size that can be reliably deployed on a laptop equipped with 16GB of RAM after applying quantization techniques, enabling a practical on-device evaluation.

For effective on-device LLM deployment, both the quantization method and the resulting data format are critical considerations. The choice of quantization method, such as symmetric or asymmetric quantization, block-wise or per-tensor quantization, directly influences the trade-off between model size reduction and potential accuracy loss. The data format resulting from quantization, which dictates the bit-width and representation of the quantized weights, significantly impacts memory footprint and computational efficiency. Thus, considering the availability in `llama.cpp`, seven quantization methods i.e., `q8_0`, `q5_0`, `q4_0`, `q5_k`, `q4_k`, `q3_k`, and `q2_k`, are adopted in this study. Table 3 details a systematic comparison of these methods across key characteristics, where $d_1$ denotes primary layer, $d_2$ denotes secondary layer, and BPW denotes the resultant bits per weight.

### 3.2 MODEL CAPABILITY

**Datasets.** We have selected five diverse benchmarks that encompass a broad spectrum of competencies. GSM8K (Cobbe et al., 2021) emphasizes multi-step reasoning by presenting grade-school mathematics problems. HellaSwag (Zellers et al., 2019) evaluates commonsense reasoning, challenging models to infer plausible continuations in everyday scenarios. Similarly, MMLU (Hendrycks et al., 2021) spans a wide array of disciplines, providing an extensive evaluation of general knowledge. Together, these benchmarks facilitate a comprehensive assessment of a model's performance, resilience, and adaptability under resource constraints. HumanEval (Chen et al., 2021) assesses the model's capabilities in generating precise and efficient code, reflecting practical software development tasks. Lastly, TruthfulQA (Lin et al., 2022) examines the reliability of factual outputs and mitigates the risks related to hallucinated information.

**Evaluation Metrics.** Evaluation metrics for the aforementioned tasks must be tailored to their unique characteristics to ensure accurate and meaningful assessment of model performance. Each task presents distinct challenges and objectives, necessitating the use of specialized metrics that align with its specific requirements. Crafting task-specific evaluation metrics not only enhances the precision of benchmarking but also captures the nuanced capabilities of language models, ensuring holistic evaluation. Specific methodologies for metric construction per dataset are detailed in Section A.4.

We adopt `lm-evaluation-harness` (Gao et al., 2024) (version `0.4.8`) as the evaluation tool to align with Open LLM Leaderboard so that all our evaluations and comparisons

are fair and clear. Given the incomplete native support of `lm-evaluation-harness` (version `0.4.8`) for `llama.cpp`, we develop a customized evaluation mechanism by extending `lm-evaluation-harness` to ensure comprehensive functionality. We keep the default settings on most hyper-parameters such as temperature and top-$p$ threshold. We set the same number of few shots according to the Qwen 2.5 technical report (Team, 2024b): 4-shots for GSM8K, 10-shots for HellaSwag, 5-shots for MMLU, and 0-shot for TruthfulQA and HumanEval. To make the evaluation results consistent, the same fewshot setting is also applied to Llama 3 models.

### 3.3 DEPLOYMENT EFFICIENCY

**Datasets.** To evaluate efficiency and sustained operational performance, we conduct text generation tasks employing extensive context sequences. These tasks utilize synthetic data generated by LLMs under evaluation. To simulate real-world application scenarios, initial inputs consist of text segments with lengths of 54, 118, 246, or 502 tokens. Each segment is prefixed with a constant 10-token (counted by `llama.cpp`) instructional prompt (*Write a 50000-word article based on this text:*) to elicit long text generation. Thus, the total input lengths are 64, 128, 256, and 512 tokens, respectively. All generation processes are uniformly terminated upon generating 1024 tokens.

**Evaluation Metrics.** The primary metric for LLM inference efficiency is throughput, measured in tokens per second. We evaluate throughput independently for the prefill (input processing) and decode (token generation) stages, defining it as the number of tokens processed divided by the execution time. For our experiments, all performance metrics are obtained directly from the runtime logs provided by the `llama.cpp` framework. We have verified the accuracy of these logs; our manual measurements of latency and throughput align with the logged values to three decimal places.

To ensure reliable performance measurements, we first pre-warm the `llama.cpp` with three preliminary runs to mitigate cold-start latency. And we fix the `temperature` and `seed` parameters to ensure reproducibility. In the event that a run fails to produce the target 1024 tokens, these parameters are adjusted, and the trial is repeated. The reported throughput is the average of three independent runs, with the KV-cache cleared before each trial to ensure measurement independence.

### 3.4 SYSTEM RESOURCE UTILIZATION

**Datasets.** Consistent with the setup in Section 3.3, we evaluate the performance of token generationtion with input lengths of 64, 128, 256, and 512 tokens.

**Evaluation Metrics.** Resource utilization is quantified by three primary metrics: CPU utilization, memory occupancy, and power consumption. Windows Performance Recorder (WPR) and Windows Performance Analyzer (WPA) are employed to systematically quantify and analyze system-level resource consumption throughout application execution, specifically monitoring CPU utilization, memory occupancy, and power draw. The `psutil` Python library is employed to quantify memory consumption. This library facilitates the retrieval of system and process information, including detailed memory usage statistics. Our analysis focuses on the Resident Set Size (RSS), representing the non-swapped physical memory utilized by the LLM inference process. The key metric adopted is Peak RSS, defined as the maximum physical RAM consumed by the process at any point during its execution, thereby capturing the worst-case memory footprint. The specific `psutil` metric for RSS on Windows corresponds to the "wset" field, aligning with the "Memory (Working Set)" in Task Manager, consistent with system-level reporting.

Energy consumption is assessed using the Windows Energy Estimation Engine (E3), an integrated real-time power modeling system. E3 translates hardware activity metrics into energy estimates for various system components. To optimize data collection and minimize extraneous recording, a customized Windows Performance Recorder (WPR) profile is developed through a structured XML configuration. This tailored profile enables targeted logging of energy-relevant counters.

## 4 EXPERIMENTS

Our experiments utilize `llama-cpp-python` (v`0.3.7`) (Andrei, 2024), an open-source Python binding for `llama.cpp`. The evaluation platform is a consumer-grade laptop with a 12-core, 2.20 GHz Intel Core i7-1360P CPU, 16 GB of 5600 MT/s DDR5 RAM, and an NVMe SSD, selected to

Table 1: Performance of selected Qwen 2.5 (Team, 2024b) instruction-tuned models.

| Models & Tasks | | Quantization | | | | | | | |
|---|---|---|---|---|---|---|---|---|---|
| | | fp16 | q8_0 | q5_k | q5_0 | q4_k | q4_0 | q3_k | q2_k |
| 14B | GSM8K (Cobbe et al., 2021) | / | / | 89.08 | 89.76 | 88.86 | 90.45 | 89.01 | 84.46 |
| | HellaSwag (Zellers et al., 2019) | / | / | 85.09 | 84.78 | 84.72 | 84.24 | 83.93 | 81.76 |
| | MMLU (Hendrycks et al., 2021) | / | / | 79.74 | 79.75 | 79.55 | 79.43 | 78.86 | 75.74 |
| | HumanEval (Chen et al., 2021) | / | / | 69.51 | 69.51 | 68.90 | 68.90 | 69.51 | 62.20 |
| | TruthfulQA (Lin et al., 2022) | / | / | 68.64 | 69.63 | 68.93 | 67.50 | 66.92 | 65.89 |
| 7B | GSM8K (Cobbe et al., 2021) | / | 86.73 | 86.05 | 87.11 | 85.52 | 86.20 | 84.46 | 76.19 |
| | HellaSwag (Zellers et al., 2019) | / | 81.32 | 81.21 | 81.19 | 80.94 | 80.59 | 79.79 | 77.48 |
| | MMLU (Hendrycks et al., 2021) | / | 74.24 | 74.27 | 74.15 | 74.26 | 74.04 | 73.23 | 68.58 |
| | HumanEval (Chen et al., 2021) | / | 70.73 | 67.68 | 68.90 | 64.63 | 58.54 | 63.41 | 53.66 |
| | TruthfulQA (Lin et al., 2022) | / | 64.74 | 64.25 | 64.45 | 63.80 | 62.22 | 64.44 | 62.25 |
| 3B | GSM8K (Cobbe et al., 2021) | 80.89 | 80.29 | 78.77 | 80.82 | 76.80 | 75.36 | 62.62 | 48.52 |
| | HellaSwag (Zellers et al., 2019) | 75.29 | 75.07 | 74.90 | 75.10 | 74.86 | 73.69 | 70.85 | 68.56 |
| | MMLU (Hendrycks et al., 2021) | 66.38 | 66.52 | 66.06 | 66.34 | 65.68 | 65.08 | 60.15 | 58.80 |
| | HumanEval (Chen et al., 2021) | 54.88 | 54.88 | 49.39 | 47.56 | 48.78 | 45.12 | 46.34 | 31.10 |
| | TruthfulQA (Lin et al., 2022) | 58.67 | 58.59 | 58.38 | 58.94 | 58.38 | 57.00 | 56.38 | 50.16 |
| 1.5B | GSM8K (Cobbe et al., 2021) | 60.80 | 59.82 | 59.14 | 57.85 | 52.77 | 53.30 | 46.47 | 21.46 |
| | HellaSwag (Zellers et al., 2019) | 67.91 | 67.92 | 67.70 | 67.66 | 66.81 | 66.89 | 65.06 | 59.92 |
| | MMLU (Hendrycks et al., 2021) | 60.28 | 60.36 | 60.32 | 59.77 | 59.76 | 59.14 | 57.33 | 51.30 |
| | HumanEval (Chen et al., 2021) | 37.20 | 37.80 | 34.76 | 39.02 | 37.50 | 35.37 | 24.39 | 20.73 |
| | TruthfulQA (Lin et al., 2022) | 46.70 | 46.70 | 45.78 | 46.41 | 45.41 | 47.71 | 45.21 | 46.51 |
| 0.5B | GSM8K (Cobbe et al., 2021) | 31.48 | 33.28 | 31.77 | 30.86 | 30.33 | 21.00 | 25.85 | 21.99 |
| | HellaSwag (Zellers et al., 2019) | 50.50 | 50.37 | 49.88 | 50.12 | 50.20 | 48.52 | 49.65 | 49.04 |
| | MMLU (Hendrycks et al., 2021) | 46.69 | 46.81 | 46.09 | 46.08 | 46.12 | 44.36 | 45.53 | 44.83 |
| | HumanEval (Chen et al., 2021) | 29.88 | 31.71 | 26.83 | 29.27 | 28.66 | 22.56 | 28.05 | 25.00 |
| | TruthfulQA (Lin et al., 2022) | 42.50 | 42.50 | 42.15 | 42.61 | 42.51 | 40.40 | 41.89 | 40.90 |

represent a typical user configuration. Given that the model's task performance is largely platform-independent (Schlögl et al., 2023), we conducted all performance evaluations on a GPU to capitalize on its significantly higher inference throughput. These evaluations were performed on a Linux workstation equipped with two 56-core Intel Xeon Platinum 8480C CPUs and an NVIDIA H800 GPU with 80 GB of VRAM.

## 4.1 MODEL CAPABILITY RESULTS

Following the experimental setup in Section 3.3, we showcase the performace of selected Qwen2.5 (Team, 2024b) intruction-tuned models, ranging in sizes from 1.5B to 14B parameters, across aforementioned tasks GSM8K (Cobbe et al., 2021), HellaSwag (Zellers et al., 2019), MMLU (Hendrycks et al., 2021), HumanEval (Chen et al., 2021), and TruthfulQA (Lin et al., 2022) in Table 1. For each model, we evaluate the original fp16 model (except 7B and 14B) with a series of quantization levels following the experimental setup introduced in Section 3.2. The results highlight the impact of both model size and quantization methods on task accuracy. From Table 1, a clear trend emerges showing that larger models in parameters generally achieve higher accuracy across almost all tasks, demonstrating their greater capacity for reasoning, knowledge retention, and adaptability. For example, the 14B models consistently outperform the smaller models, with different quantization series yielding top scores on GSM8K (Cobbe et al., 2021) (q4_0), HellaSwag (Zellers et al., 2019) (q5_k), MMLU (Hendrycks et al., 2021) (q5_0), and TruthfulQA (Lin et al., 2022) (q5_0). Smaller models, such as the 3B and 1.5B variants, show a noticeable decline in accuracy, particularly on reasoning-intensive tasks like GSM8K and MMLU. It is notable that the 7B models, particularly the q8_0 model, achieve results on HumanEval (Chen et al., 2021) that are comparable to, and even slightly better than, some of the 14B models. While this might seem counterintuitive at first, such behavior is also observed in (Team, 2024b).

Within each model size, the choice of quantization has a substantial impact on performance. The fp16 model generally achieves the highest accuracy, suggesting that lower quantization levels may degrade performance due to precision loss. Additionally, the results reveal that larger models are more robust to variations in quantization levels compared to smaller models. For example, the 14B model on GSM8K (Cobbe et al., 2021) demonstrates only a slight reduction in accuracy, from 89.08% (q5_k) to 89.01% (q3_k). In contrast, the 1.5B model shows a more pronounced decline, with its accuracy decreasing from 59.14% (q5_k) to 46.47% (q3_k). The findings from Table 1 indicate that larger models are more resilient to performance loss caused by lower-precision quantization. However,

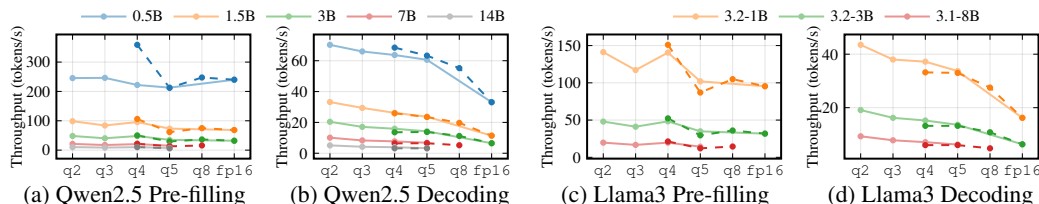

Figure 1: Pre-filling and decoding throughput of selected models of Qwen 2.5 (Team, 2024b) and Llama 3 (Team, 2024a) against different quantization methods (128-token input).

the `q2_k` configuration significantly degrades performance across all model sizes, demonstrating that extreme quantization harms both computational accuracy and representation fidelity. These results highlight the robustness of larger models, likely due to their higher parameter capacity, and offer insights for optimizing quantization in smaller models to ensure usability in resource-limited environments, such as on-device LLMs.

Moreover, task-specific performance provides insights into the models' strengths and weaknesses. Tasks like HumanEval (Chen et al., 2021), which involve code generation, exhibit greater sensitivity to model size and quantization. For example, the accuracy for HumanEval drops sharply in smaller models, with the 1.5B `q5_0` model achieving 39.02% compared to 69.51% in the 14B counterpart. TruthfulQA (Lin et al., 2022) similarly shows a decline in factual reliability as model size decreases, highlighting the challenges of maintaining accuracy in smaller models.

## 4.2 DEPLOYMENT EFFICIENCY RESULTS

We showcase the pre-filling and decoding efficiencies of selected instruction-tuned models of Qwen 2.5 (Team, 2024b) and Llama 3 (Team, 2024a) across the quantization series introduced in Section 3.1. We highlight the impact of model size, prompt length, and quantization methods on efficiency.

Figure 1 depicts throughput as a function of model size with 128 input tokens. Solid bold curves correspond to the `n_k` quantization family, while dashed curves indicate the `n_0` family. Individual models, detailed in Section 3.1, are distinguished by color. Two primary trends are evident: 1) A monotonic decline in throughput is observed with increasing model scale. 2) The choice of quantization method exerts a diminishing influence at larger model scales. These findings suggest that when deploying LLMs on resource-constrained devices, higher BPW quantization methods offer a balance between accuracy and computational cost, preserving accuracy with a minimal latency penalty.

**Impact of Quantization Method.** Moreover, we analyze the impact of quantization technique on throughput. During decoding (Figures 1b and 1d), throughput declines monotonically with increasing BPW, as higher-precision quantization methods entail greater computational and memory traffic overhead. This effect is considerably more pronounced than during the pre-filling phase (Figures 1a and 1c), consistent with decoding being predominantly memory-bound. In addition, operator-level optimizations within `llama.cpp` amplify the performance advantage of the `n_k` series; For 4-bit and 5-bit configurations, variants `n_k` outperform their counterparts `n_0`.

During the pre-filling phase (Figures 1a and 1c), throughput generally decreases as BPW increases. However, since pre-filling is compute-bound, the impact of CPU computational overhead is magnified relative to the decoding phase. Specifically, the `q5_k` quantization scheme achieves a greater speedup relative to `q5_0` due to its integration of 6-bit quantization within its 5-bit framework. Both methods partition quantized weights across multiple bytes. The 6-bit quantization component within `q5_k` requires fewer data concatenation iterations to form 8-bit or 16-bit data units during the unpacking process (marcingomulkiewicz, 2024). Conversely, this mixed-precision approach, characteristic of `n_k` variants, introduces additional bit-shifting operations. For instance, while the `q4_0` method employs a single shifting operation to unpack quantized weights into an 8-bit data unit, the more complex unpacking logic in `q4_k` (and other `_k` schemes) may result in slightly reduced throughput compared to the simpler `q4_0`. `q2_k` and `q3_k` necessitate even more shifting operations and iterations, leading to lower throughput compared to 4-bit methods (`q4_k` and `q4_0`). However, for very small models, such as Qwen 2.5 0.5B (Figure 1a), low-bit quantization can improve cache hit rates, thereby enhancing throughput. Thus, in such cases, the increased computational overhead associated with complex unpacking can be offset by gains in memory access efficiency.

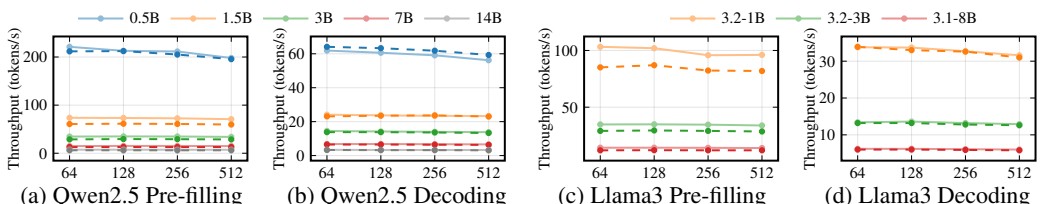

(a) Qwen2.5 Pre-filling  (b) Qwen2.5 Decoding  (c) Llama3 Pre-filling  (d) Llama3 Decoding

Figure 2: Pre-filling and decoding throughput with selected models of Qwen 2.5 (Team, 2024b) and Llama 3 (Team, 2024a) against different token lengths (5-bit quantization).

**Impact of Token Length.** Figure 2 illustrates throughput performance as a function of model size under 5-bit quantization schemes (`q5_k` and `q5_0`). During decoding, a fixed output length of 1024 tokens is employed. The observed marginal decrease (3% on 14B and 7B, 5% on 3B) in throughput with increasing token length indicates minimal growth in communication or computation overheads, suggesting neither currently constitutes the primary bottleneck. Regarding communication, model weight transmission overhead is dominant for shorter token sequences, as the incremental communication cost of the KV-cache is negligible relative to that of the model weights. Conversely, for smaller models, an increasing proportion of KV-cache communication, relative to model weight transmission, results in a more significant throughput degradation (7% on 1B, 9% on 0.5B).

As illustrated in Figure 1, the negative impact of BPW on throughput is less pronounced in smaller models. This trend is particularly evident when analyzing the performance degradation from low precision (`q2`) to high precision (`fp16`), with data presented in Figures 1a and 1b. During the computation-intensive prefill stage, increasing the BPW results in a throughput degradation of only 2.3% for the Qwen2.5 0.5B model, compared to a substantial 30.5% for the larger Qwen2.5 1.5B model. This wide disparity suggests that as model size decreases, the primary performance bottleneck shifts from computation to communication.

The communication-dominance hypothesis is further nuanced by the results in Figure 2, where longer input token lengths mitigate the overall throughput degradation from higher Bits Per Weight (BPW). For example, when increasing the BPW from `q2` to `fp16` (data in Figures 2a and 2b), the throughput degradation for the Qwen2.5 0.5B model is 10.5% (prefill) and 9.2% (decode). In contrast, the larger Qwen2.5 1.5B model is less affected, with degradation ratios of 3.9% and 3.3%, respectively. Consequently, the primary bottleneck is contingent on both model scale and workload. This is particularly salient for deployments on edge devices, where the computational load on the CPU must be balanced against inherent communication constraints.

### 4.3 SYSTEM RESOURCE UTILIZATION RESULTS

An analysis of CPU power consumption for quantized LLMs on edge devices reveals a largely stable power draw across various bit-widths (`q2_k` to `q8_0`), with values fluctuating within a narrow 7.9W–9.5W range (Figure 3a). However, subtle inefficiencies in mixed-precision processing are observed, with `q4_0` (9.2W) and `q8_0` (9.5W) consuming marginally more power than `q5_0` (8.5W). We attribute this to hardware-level optimizations; for example, the alignment of `q5_0` with 32-bit registers likely minimizes instruction pipeline stalls, while the irregular bit grouping in `q4_0` may incur additional overhead from bit-unpacking operations.

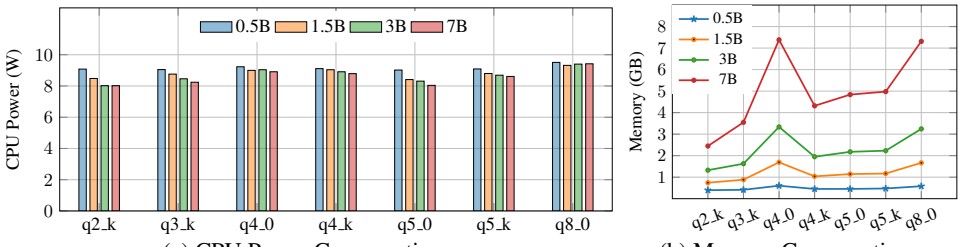

(a) CPU Power Consumption.  (b) Memory Consumption.

Figure 3: System resource utilization (power in Watt and memory in GB) of Qwen 2.5 series (from 0.5B to 7B). Prompt size of 128 tokens and output size 1000 tokens are fixed.

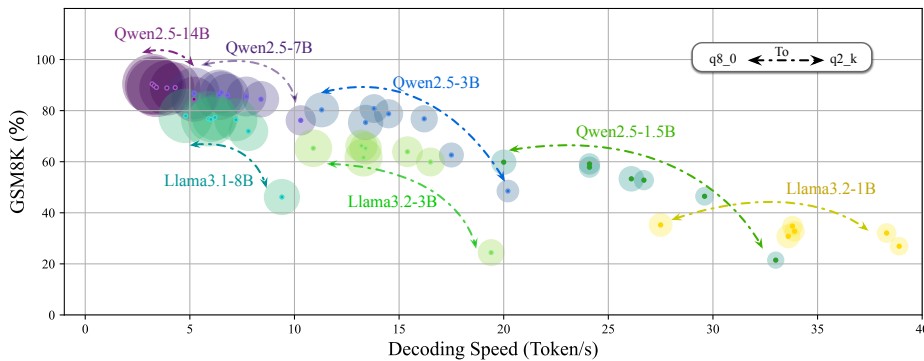

Figure 4: Visualiszation of decoding speed (token/s) and model performance (GSM8K score). Llama 3 and Qwen 2.5 series are evaluated at different quantization scheme (`q8_0` to `q2_k`). The circle area of each model is proportional to its memory consumption on device.

Quantization bit-width fundamentally alters the relationship between model scale and power consumption (Figure 3a). At `q2_k` precision, we observe a paradoxical trend where larger models consume less power; for example, the 7B model uses 8.27W versus 9.33W for the 0.5B model. This occurs because the workload shifts from being compute-bound to memory-bound at this low precision. As a result, CPU power draw is dictated by the latency of data movement from memory—during which the CPU is often underutilized—rather than by the model's computational complexity.

However, the above trend disappears at `q8_0`, where the 7B and 0.5B models exhibit minimal divergence (9.31W vs. 9.65W, ↑0.34W). We observe two key patterns: 1) Ultra-low bit quantization (`q2_k`) enables memory access optimization for larger models (e.g., reduced cache misses due to full weight residency), overriding computational load increases; 2) High-bit operations (`q8_0`) saturate CPU arithmetic units regardless of model scale, diminishing size-related efficiency variations. The results suggest that aggressive quantization shifts power bottlenecks from computation to memory subsystems, with diminishing returns as bit-width increases.

Moreover, memory consumption scales near-monotonically with quantization bit-width for all schemes except `q4_0`. For example, 0.5B models range from 392 MB (`q2_k`) to 576 MB (`q8_0`), and 7B models from 2411 MB to 7297 MB. The `q4_0` scheme consistently breaks this trend, exhibiting anomalously high memory usage across all model scales; the 0.5B variant, for instance, requires 597 MB (versus 448 MB for `q5_0`). This is caused by runtime weight repacking, an optimization specific to the `q4_0` implementation in `llama.cpp`. To enhance computational performance, this process reorganizes 4-bit weight groups into 32-bit aligned blocks, a design that prioritizes execution speed over memory efficiency. Apart from this specific implementation detail, memory usage for other methods scales predictably in proportion to bit-width.

Our analysis indicates that CPU power consumption is fundamentally governed by the balance between computation-intensive and memory-intensive workloads, not just high-level model properties. Models that are compute-bound consume significantly more power than those that become memory-bound, as the latter leads to reduced CPU utilization and frequency. This principle is evident across various quantization schemes (Figures 3 and 13). High-power models such as `q4_k`, `q4_0`, and `q8_0` sustain high CPU utilization, particularly during decoding. In contrast, low-bit-width models like `q2_k`, `q3_k`, and `q5_0` become memory-bound, leading to lower power consumption manifested through reduced CPU utilization and increased System Agent (SA) power. This trade-off also explains specific behaviors: `q4_0` consumes more power than `q4_k` due to a compute-intensive prefill phase, while `q5_0` uses less power than `q5_k` by inducing a memory-intensive decoding phase. Therefore, we conclude that low-level operational characteristics dictated by the quantization implementation are the primary determinant of CPU power consumption.

Further, we evaluate the interplay between accuracy, speed, and memory consumption, as illustrated by the Pareto frontier analysis in Figure 4. This analysis reveals three key findings. First, Qwen 2.5 models consistently outperform Llama variants at all scales. Quantization to 4-bit precision offers a favorable trade-off, accelerating inference by 30-50% with minimal accuracy loss, whereas further compression below 4-bits leads to severe degradation. Second, robustness to quantization is

scale-dependent. The Qwen2.5 7B model, for instance, maintains accuracy comparable to its 14B counterpart at double the speed, while sub-3B models suffer disproportionate accuracy losses when quantized. Moreover, and most critically, model scale is a more dominant factor than quantization in defining the efficiency-accuracy trade-off. Scaling a model from 7B to 14B parameters sacrifices significant speed for accuracy, while 4-bit quantization on the 7B model provides a substantial speedup with a negligible accuracy drop. This establishes model scaling as the primary determinant of the overall performance balance, with quantization serving as a secondary tool for fine-tuning speed and memory.

## 5 CONCLUSION AND DISCUSSION

This paper presents a systematic evaluation of LLM inference on edge devices. Our experimental results yield several key findings concerning model capability, deployment efficiency, and system resource utilization across diverse model sizes and quantization methods.

**Model Capability.** Model capability increases monotonically with BPW and model scale. Qwen models generally outperform Llama models. Further, large models employing low-bit quantization demonstrate superior performance compared to smaller models utilizing higher-bit quantization.

**Deployment Efficiency.** LLM inference throughput is primarily governed by an inverse correlation with model size and Bits Per Weight (BPW). During the communication-bound decoding phase, higher BPW typically results in a monotonic decrease in throughput. However, this trend is nuanced by the computational cost of the specific quantization method. For a fixed bit-width, the efficiency of the bit-unpacking operation—where methods requiring fewer discrete operations like bit-shifting yield higher performance—emerges as a critical factor. Furthermore, for sufficiently small models, communication bottlenecks can be alleviated; their reduced model and KV-cache footprints can improve cache hit rates and increase throughput, counteracting the general scaling trends. During the pre-filling phase, the dense computations inherent to batch processing typically render the system computation-bound, particularly for large models. Consequently, the computational efficiency of the selected quantization method becomes paramount for throughput in these scenarios. Conversely, for small models during pre-filling, which are generally less computation-bound, effective management of communication overheads—facilitated by smaller model and KV-cache sizes—is crucial for optimizing throughput.

**System Resource Utilization.** Memory consumption exhibits a monotonic increase with BPW. Quantization methods characterized by fewer CPU operations during the unpacking stage generally correlate with higher overall CPU utilization, as computational throughput is less frequently impeded by this process compared to less efficient alternatives.

The observations lead to the following insights, offered as recommendations for model and quantization method selection, particularly for on-device framework development and efficiency optimization:

**Trade-off between Model Capability and Deployment Efficiency**. For large-scale models, low-bit-width quantization typically preserves accuracy while offering only marginal gains in deployment efficiency. Conversely, for small-scale models, low-bit-width quantization achieves accuracy comparable to its high-bit-width counterpart while substantially improving deployment efficiency.

**Model Selection for Resource-Constrained Scenarios:** In scenarios prioritizing accuracy, the deployment of *large models with moderate quantization precision* (e.g., 4-bit) is often advisable, as this frequently represents an optimal balance among capability, efficiency, and resource consumption. Conversely, in scenarios where deployment efficiency is paramount, employing *small models, also with moderate quantization precision,* can be more effective.

**Bottleneck Identification on Edge Devices:** On edge devices, the primary performance bottleneck is highly dependent on model scale. For larger models (>1B parameters), the limited parallelism of CPUs renders them computation-bound, even during decoding. Consequently, optimizing computational efficiency is the most effective strategy for improving throughput. In contrast, for smaller models (<0.5B parameters) in long-context input scenarios, communication overhead potentially becomes the dominant constraint. Therefore, enhancing data transfer efficiency yields more substantial performance gains in this regime.

## 6 REPRODUCIBILITY STATEMENT

This paper is reproducible. The code is open-source. The detailed proofs of theoretical results are available in the Appendix.

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

# A APPENDIX

## A.1 COMPARISON OF DIFFERENT LLM INFERENCE FRAMEWORKS

We survey existing LLM inference frameworks for a CPU-based Windows platform. The results are shown in Table 2. Specifically, frameworks such as TensorRT-LLM, SGLang, and exllama are incompatible as they are designed for GPU architectures. Although we deployed vLLM via WSL, it demonstrated unacceptably low throughput (e.g., 3.07 tokens/s for a 0.5B model), encountered out-of-memory errors with models larger than 1.5B, and lacked `int4`/`int8` quantization support on its CPU backend. MLC-LLM also performed poorly, yielding extremely slow inference that stalled after a few tokens. This lack of viable alternatives led us to select `llama.cpp` to investigate the performance limits of LLMs on consumer hardware, a choice justified by its role as a foundational backend for many popular tools like Ollama.

Table 2: Comparison of Different LLM Inference Frameworks.

| Inference Framework | Platforms | Limitations |
|---|---|---|
| vLLM | GPUs / Server-level CPUs | Memory hungry. Lack `int4` and `int8` quantization support on CPU. Slow on CPU. |
| TensorRT-LLM | NVIDIA GPUs only | Not for CPUs. |
| SGlang | GPUs / Server-level CPUs | Lack support for consumer CPUs without AMX Instructions. |
| exllamav2/v3 | Single GPU | Only support devices with CUDA. |
| MLC-LLM | GPU/CPU/Mobile CPU/NPU | Extremely slow on consumer CPU. Vulkan backend, limits memory to 6GB. |

## A.2 DETAILS OF IMPORTANCE MATRIX

To reduce the precision degradation between de-quantized and pre-quantized values, `llama.cpp` (ggml-org, 2024) formulates an optimization problem that leverages activation statistics to construct weight-aware objective functions. Let $\mathbf{w} \in \mathbb{R}^d$ denote a $d$-dimensional learnable parameter vector, with $\mathbf{q} \in \mathbb{Q}^d$ representing its quantized counterpart obtained through standard quantization procedures. Let $\mathbf{a} \in \mathbb{R}^d$ denotes the corresponding activation vector from the preceding layer. For each dimension $i \in \{1, 2, \ldots, d\}$, the per-block quantization problem can be formulated as a quadratic programming problem as $\min_{s,m} \mathbb{E}\big[ \sum_{i=1}^d (\mathbf{q}_i - \mathbf{w}_i)\mathbf{a}_i \big]^2$ ($s$ and $m$ are the scale factor and the minimum value defined in Section 2). Taking the reformulation in (DavidZyy, 2024), the sum-squared operation in the objective above can be simplified into squared-sum form as:

$$\mathbb{E}\left[ \sum_{i=1}^d (\mathbf{q}_i - \mathbf{w}_i)\mathbf{a}_i \right]^2 \approx \mathbb{E}\left[ \sum_{i=1}^d \mathbf{a}_i^2 (\mathbf{q}_i - \mathbf{w}_i)^2 \right], \tag{1}$$

where the coefficient $\mathbf{a}_i$ serves as a weighting factor in the optimization objective.

*Proof.*

$$\mathbb{E}\left[\sum_{i=1}^{d}(\mathbf{q}_i - \mathbf{w}_i)\mathbf{a}_i\right]^2$$

$$\text{Expansion of quadratic} \implies = \mathbb{E}\left[\sum_{i=1}^{d}\left((\mathbf{q}_i - \mathbf{w}_i)\mathbf{a}_i\right)^2 + \sum_{j=1, j\neq i}^{d}(\mathbf{q}_i - \mathbf{w}_i)\mathbf{a}_i(\mathbf{q}_j - \mathbf{w}_j)\mathbf{a}_j\right],$$

$$\text{Sum of expectation} \implies = \mathbb{E}\left[\sum_{i=1}^{d}\left((\mathbf{q}_i - \mathbf{w}_i)\mathbf{a}_i\right)^2\right] + \mathbb{E}\left[\sum_{j=1, j\neq i}^{d}(\mathbf{q}_i - \mathbf{w}_i)\mathbf{a}_i(\mathbf{q}_j - \mathbf{w}_j)\mathbf{a}_j\right],$$

$$\text{Eliminate Second Term} \implies \approx \mathbb{E}\left[\sum_{i=1}^{d}\mathbf{a}_i^2(\mathbf{q}_i - \mathbf{w}_i)^2\right].$$

$\square$

Empirical studies suggest that parameters with larger magnitudes exhibit greater influence in neural network computations (jukofyork, 2024). To amplify the significance of these salient parameters, a squared magnitude term $x_i^2$ is incorporated into the weighting factor (jukofyork, 2024). $x_i^2$ also serves as a simple approximation of the diagonal entries of Hessian matrices (jukofyork, 2024). Moreover, to address numerical instability in low-magnitude regimes, block-wise mean squared value of original data is calculated as $\sigma_2 = \frac{1}{n}\sum_{i=1}^{n}\mathbf{x}_i^2$ (jukofyork, 2024). This regularization prevents the systematic underestimation of near-zero parameters during quantization (jukofyork, 2024). The complete importance matrix is therefore formulated as $\tilde{\mathbf{a}}_i^2 = \mathbf{a}_i^2\sqrt{\sigma_2 + \mathbf{x}_i^2}$.

## A.3 DETAILS OF SELECTED QUANTIZATION METHODS OF LLAMA.CPP (GGML-ORG, 2023)

We begin by introducing the *mini-block technique* utilized in llama.cpp, which enhances data representation fidelity by quantizing parameters into smaller, independent groups. LLM parameters are partitioned into contiguous blocks $\mathcal{B}_i$, each of a predefined size $d_1 \in \mathbb{N}^+$. Each block $\mathcal{B}_i$ is then quantized independently using its own scale factor $s_i$ and zero-point (denoted as $z_i$). Thus, for any parameter $w \in \mathcal{B}_i$, its quantized value $q_w$ is given by $q_w = \mathcal{Q}(w; s_i, z_i)$, where $\mathcal{Q}(\cdot)$ is the quantization function. This allows for finer-grained adaptation to the local data distribution within each block.

The q8_0, q5_0, and q4_0 methods employ symmetric quantization with a single-layer block structure, grouping 32 weights per block. These methods allocate 8, 5, and 4 bits to weight representation, respectively, while uniformly reserving 16 bits to store quantization parameters (scale and minimum values). Due to the additional bit overhead for shared scale parameters, the resulting BPW for q8_0, q5_0, and q4_0 are calculated as 8.5, 5.5, and 4.5 bits, respectively.

The remaining n_k quantization methods employ distinct precision-mixed strategies tailored to specific model components, featuring diverse block sizes and bit allocations for quantization parameters. As detailed in Table 3, the q5_k method utilizes symmetric quantization for half of the "attention.wv" and "feed_forward.w2" components, structured with a primary block (16 weights) and a secondary block (16 parameters). Here, weights and scale parameters are allocated 6 bits and 8 bits, respectively, resulting in a BPW of 6.5625. For asymmetric quantization, q5_k employs a 32×8 block configuration with 5 bits for weights and 6 bits for parameters, yielding a BPW of 5.5. The component-specific configurations in q4_k mirror those of q5_k, but with reduced bit allocations: 4 bits for weights and 6 bits for scale parameters, achieving a lower BPW of 4.5.

In the q3_k method, components attention.wv, attention.wo, and feed_forward.w2 utilize a 32×8 block configuration, allocating 4 bits for weights and 6 bits for scale parameters, yielding 4.5 BPW. For other components, weights are quantized using a 16×16 block structure with 3 bits for weights and 6 bits for scale parameters, achieving a reduced BPW of 3.4375. The q2_k method adopts component-specific configurations analogous to q4_k and q5_k, but with distinct parameterizations. For attention.wv and feed_forward.w2, a 32×8 block is employed with 4 bits for weights and 6 bits for scale parameters. The remaining components leverage a 16×16

Table 3: Selected quantization methods of `llama.cpp`

| Series | Weight Component | Symmetric | $d_1$ | $d_2$ | Bits | $s, m$ Bits | BPW |
|---|---|---|---|---|---|---|---|
| q8_0 | All | ✓ | 32 | N/A | 8 | 16 | 8.5 |
| q5_0 | All | ✓ | 32 | N/A | 5 | 16 | 5.5 |
| q4_0 | All | ✓ | 32 | N/A | 4 | 16 | 4.5 |
| q5_k | attention.wv: Half | ✓ | 16 | 16 | 6 | 8 | 6.5625 |
| | feed_forward.w2: Half | ✓ | 16 | 16 | 6 | 8 | 6.5625 |
| | Others | ✗ | 32 | 8 | 5 | 6 | 5.5 |
| q4_k | attention.wv: Half | ✓ | 16 | 16 | 6 | 8 | 6.5625 |
| | feed_forward.w2: Half | ✓ | 16 | 16 | 6 | 8 | 6.5625 |
| | Others | ✗ | 32 | 8 | 4 | 6 | 4.5 |
| q3_k | attention.wv: All | ✗ | 32 | 8 | 4 | 6 | 4.5 |
| | attention.wo: All | ✗ | 32 | 8 | 4 | 6 | 4.5 |
| | feed_forward.w2: All | ✗ | 32 | 8 | 4 | 6 | 4.5 |
| | Others | ✓ | 16 | 16 | 3 | 6 | 3.4375 |
| q2_k | attention.wv: All | ✗ | 32 | 8 | 4 | 6 | 4.5 |
| | feed_forward.w2: All | ✗ | 32 | 8 | 4 | 6 | 4.5 |
| | Others | ✗ | 16 | 16 | 2 | 4 | 2.5625 |

block structure, allocating 2 bits for weights and 4 bits for scale parameters, further optimizing the BPW metric.

## A.4 DETAILS OF METRIC CONSTRUCTION FOR MODEL CAPABILITY

For GSM8K, there are two types of matching scores: `flexible-extract` and `strict-match`. The former retrieves the last number in the response as the final answer to a mathematical question, disregarding the exact pattern of the answer. The latter one strictly matches the first number with a given pattern in the response. We use the maximum number between `flexible-extract` and `strict-match` matching scores as the evaluation metric.

For HumanEval, the model's code completion ability is assessed using the `pass@1` metric. It is given a function signature and a docstring describing the function, then tasked with completing it. The pass rate is determined by counting how many generated code snippets pass all test cases.

HellaSwag, MMLU, and TruthfulQA are all multiple-choice evaluation tasks. Accuracy is determined by evaluating the model's ability to select the correct answer from a set of predefined options. In HellaSwag and MMLU, model capabilities are examined by presenting contextual information without explicit choices, utilizing the logits generated during inference to compute the cumulative log probabilities for each candidate response. For TruthfulQA, we adopt the `mc2` score to measure both truthfulness and informativeness, enabling the model to assign probability values to multiple correct answers.

## A.5 ADDITIONAL RESULTS OF MODEL CAPABILITY

We also showcase the performace of selected Llama 3 intruction-tuned models, ranging in sizes from 1B to 8B parameters, across aforementioned tasks GSM8K, HellaSwag, MMLU, HumanEval, and TruthfulQA in Table 4. Similar to the observations on Qwen 2.5 models, we conclude that larger models in Llama 3 demonstrate superior performance across all tasks compared to smaller models. For example, on GSM8K, the Llama 3.1 8B `fp16` model achieves 78.01% accuracy, while the Llama 3.2 1B `fp16` model reaches only 35.03%. Similar trends are observed across other benchmarks, reflecting the general advantage of model size in capturing and utilizing complex knowledge.

Within each model size, quantization levels significantly impact performance. Lower quantization levels (e.g. `q3_k`, `q2_k`) lead to noticeable performance declines. For instance, the Llama 3.1 8B

Table 4: Performance of selected Llama 3 (Team, 2024a) instruction-tuned models.

| Models & Tasks | | fp16 | q8_0 | q5_k | q5_0 | q4_k | q4_0 | q3_k | q2_k |
|---|---|---|---|---|---|---|---|---|---|
| | | | | | **Quantization** | | | | |
| 3.1-8B | GSM8K (Cobbe et al., 2021) | / | 77.94 | 77.33 | 76.57 | 76.42 | 76.80 | 71.95 | 46.17 |
| | HellaSwag (Zellers et al., 2019) | / | 80.49 | 80.39 | 80.19 | 79.85 | 79.94 | 78.93 | 77.37 |
| | MMLU (Hendrycks et al., 2021) | / | 68.42 | 68.25 | 68.22 | 67.57 | 66.99 | 66.47 | 59.46 |
| | HumanEval (Chen et al., 2021) | / | 62.80 | 63.41 | 62.20 | 61.59 | 61.59 | 61.59 | 43.29 |
| | TruthfulQA (Lin et al., 2022) | / | 54.40 | 54.09 | 54.42 | 53.53 | 52.13 | 52.78 | 45.68 |
| 3.2-3B | GSM8K (Cobbe et al., 2021) | 64.90 | 65.28 | 65.28 | 66.26 | 63.91 | 61.64 | 59.89 | 24.41 |
| | HellaSwag (Zellers et al., 2019) | 73.62 | 73.55 | 73.43 | 73.20 | 72.60 | 72.86 | 70.93 | 61.29 |
| | MMLU (Hendrycks et al., 2021) | 60.83 | 60.75 | 60.44 | 60.30 | 60.08 | 59.72 | 57.09 | 46.52 |
| | HumanEval (Chen et al., 2021) | 50.61 | 48.78 | 50.61 | 49.39 | 51.22 | 50.61 | 47.56 | 26.83 |
| | TruthfulQA (Lin et al., 2022) | 51.46 | 51.66 | 51.82 | 50.91 | 52.09 | 50.26 | 66.34 | 45.85 |
| 3.2-1B | GSM8K (Cobbe et al., 2021) | 35.03 | 35.25 | 34.80 | 32.75 | 32.07 | 30.78 | 26.91 | 2.96 |
| | HellaSwag (Zellers et al., 2019) | 60.94 | 61.00 | 60.76 | 60.98 | 60.04 | 58.94 | 58.53 | 45.66 |
| | MMLU (Hendrycks et al., 2021) | 46.27 | 46.35 | 45.91 | 45.94 | 44.29 | 44.40 | 43.11 | 31.35 |
| | HumanEval (Chen et al., 2021) | 34.15 | 34.15 | 35.98 | 32.93 | 31.10 | 29.27 | 26.83 | 4.88 |
| | TruthfulQA (Lin et al., 2022) | 43.39 | 43.52 | 43.43 | 43.47 | 44.24 | 43.51 | 40.86 | 42.48 |

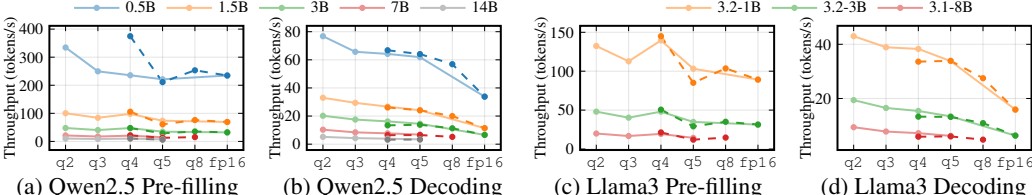

| (a) Qwen2.5 Pre-filling | (b) Qwen2.5 Decoding | (c) Llama3 Pre-filling | (d) Llama3 Decoding |

Figure 5: Pre-filling and decoding throughput of selected models of Qwen 2.5 (Team, 2024b) and Llama 3 (Team, 2024a) against different quantization methods (64-token input).

model achieves 68.26% on MMLU with `fp16` but drops to 66.47% with `q3_k` and further plummets to 59.46% with `q2_k`. Similarly, for the Llama 3.2-1B model, accuracy on MMLU decreases from 46.27% (`fp16`) to 43.11% (`q3_k`) and sharply to 31.35% (`q2_k`). The drop in quality at `q2_k` is particularly pronounced, emphasizing the limitations of extreme quantization.

Both Llama 3 and Qwen 2.5 consistently demonstrate that larger models outperform smaller ones across nearly all tasks. However, Qwen 2.5's (Team, 2024b) 7B models surpass Llama 3 8B models on all five tasks, despite having a comparable number of parameters. Similar to Qwen 2.5, Llama 3 maintains stable performance with moderate quantization levels (e.g., `q8_0`, `q5_k`), but its accuracy degrades significantly under `q2_k`. Qwen 2.5 models exhibit better resilience to quantization changes compared to Llama 3. For example, the Qwen 2.5 14B model on MMLU shows only a minor drop from 79.74% (`q5_k`) to 78.86% (`q3_k`), whereas the Llama 3.1 8B model experiences a sharper decline from 68.25% (`q5_k`) to 66.47% (`q3_k`). Furthermore, it is noticeable that the quantization `q2_k` of the Llama 3.2 1B model proves to be significantly ineffective for tasks such as GSM8K and HumanEval.

## A.6 ADDITIONAL RESULTS OF DEPLOYMENT EFFICIENCY

The subsequent figures present further throughput results of LLMs with different quantization methods, employing selected quantization methods across input token lengths of 64, 256, and 512. The results consistently reveal a degradation in decoding throughput with increasing BPW, alongside variations in pre-filling throughput linked to the operational complexity inherent in the quantization methods.

Further, the subsequent figures present additional results for LLMs across token lengths of 64, 256, and 512, employing quantization at 2-bit, 3-bit, 4-bit, and 8-bit precision levels. These results consistently reveal an exacerbated trend of throughput degradation with increasing model size, further underscoring the growing impact of computational bottlenecks at larger model scales. Notably, among

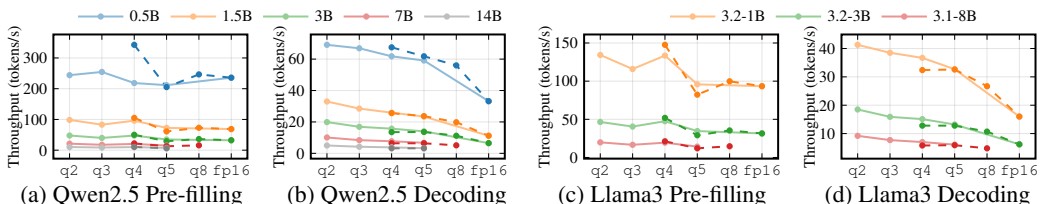

Figure 6: Pre-filling and decoding throughput of selected models of Qwen 2.5 Team (2024b) and Llama 3 Team (2024a) against different quantization methods (256-token input).

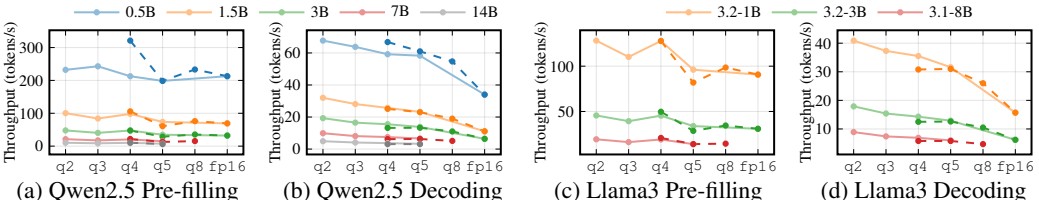

Figure 7: Pre-filling and decoding throughput of selected models of Qwen 2.5 (Team, 2024b) and Llama 3 (Team, 2024a) against different quantization methods (512-token input).

4-bit quantization methods, q4_0 exhibits superior throughput compared to q4_k, an advantage attributed to its implementation involving fewer CPU operations.

### A.7 ADDITIONAL RESULTS OF SYSTEM RESOURCE UTILIZATION

To facilitate a detailed analysis of system resource utilization, additional metrics are collected using HWiNFO64. Following a warm-up phase utilizing q2_k quantization, inference scripts are sequentially executed on the Qwen 2.5 7B and Llama 3.1 8B models. These models are evaluated with a suite of quantization schemes: q2_k, q3_k, q4_k, q5_k, q4_0, q5_0, and q8_0. Each model instance underwent approximately 10 minutes of inference, employing 256 input tokens and generating 1000 output tokens.

Metrics, including CPU utilization, CPU frequency, and System Agent (SA) Power, are sampled using HWiNFO64. Key observations from this analysis are as follows: The pre-filling phase consistently exhibits significantly higher instantaneous power consumption compared to the decoding phase; however, average power consumption is predominantly dictated by the longer decoding phase. SA Power, an integral power management component within Intel CPU architectures responsible for regulating elements such as memory controllers, PCIe controllers, and display engines, effectively reflects memory access intensity and bandwidth utilization. Consequently, elevated SA Power levels observed with the q2_k, q3_k, and q5_0 quantization schemes are indicative of memory-bound workloads.

## B THE USE OF LARGE LANGUAGE MODELS STATEMENT

The use of Large Language Models in this work was restricted to polishing writing.

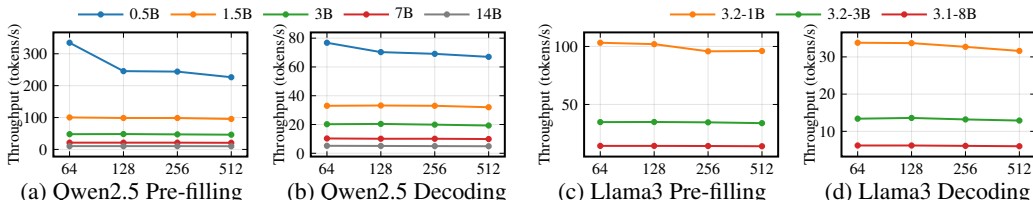

Figure 8: Pre-filling and decoding throughput with selected models of Qwen 2.5 (Team, 2024b) and Llama 3 (Team, 2024a) against different token lengths (`q2_k` quantization).

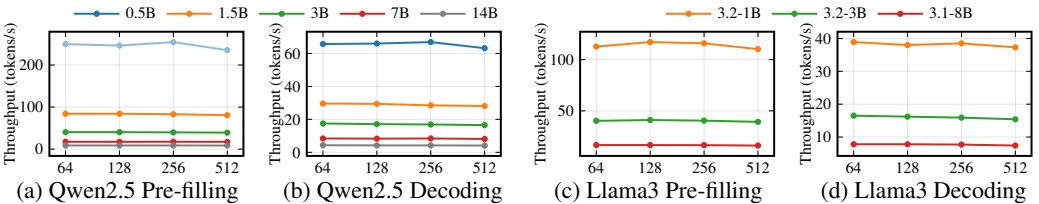

Figure 9: Pre-filling and decoding throughput with selected models of Qwen 2.5 (Team, 2024b) and Llama 3 (Team, 2024a) against different token lengths (`q3_k` quantization).

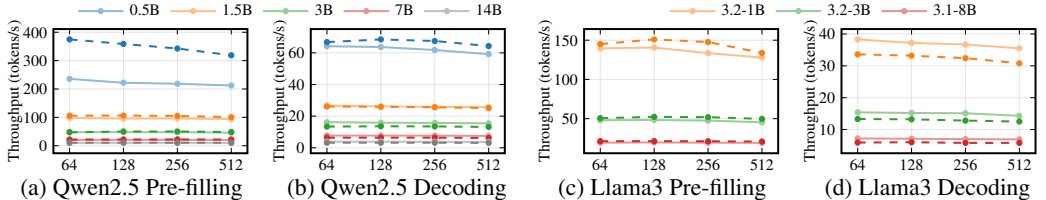

Figure 10: Pre-filling and decoding throughput with selected models of Qwen 2.5 (Team, 2024b) and Llama 3 (Team, 2024a) against different token lengths (4-bit quantization).

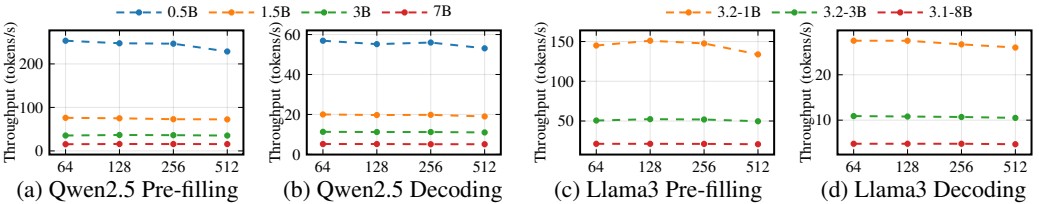

Figure 11: Pre-filling and decoding throughput with selected models of Qwen 2.5 (Team, 2024b) and Llama 3 (Team, 2024a) against different token lengths (`q8_0` quantization).

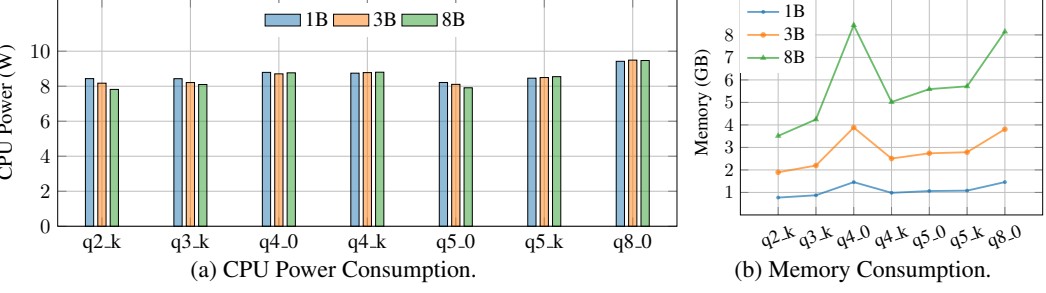

Figure 12: System resource utilization (power in Watt and memory in GB) of Llama 3 series (from 1B to 8B). Prompt size of 128 tokens and output size 1000 tokens are fixed.

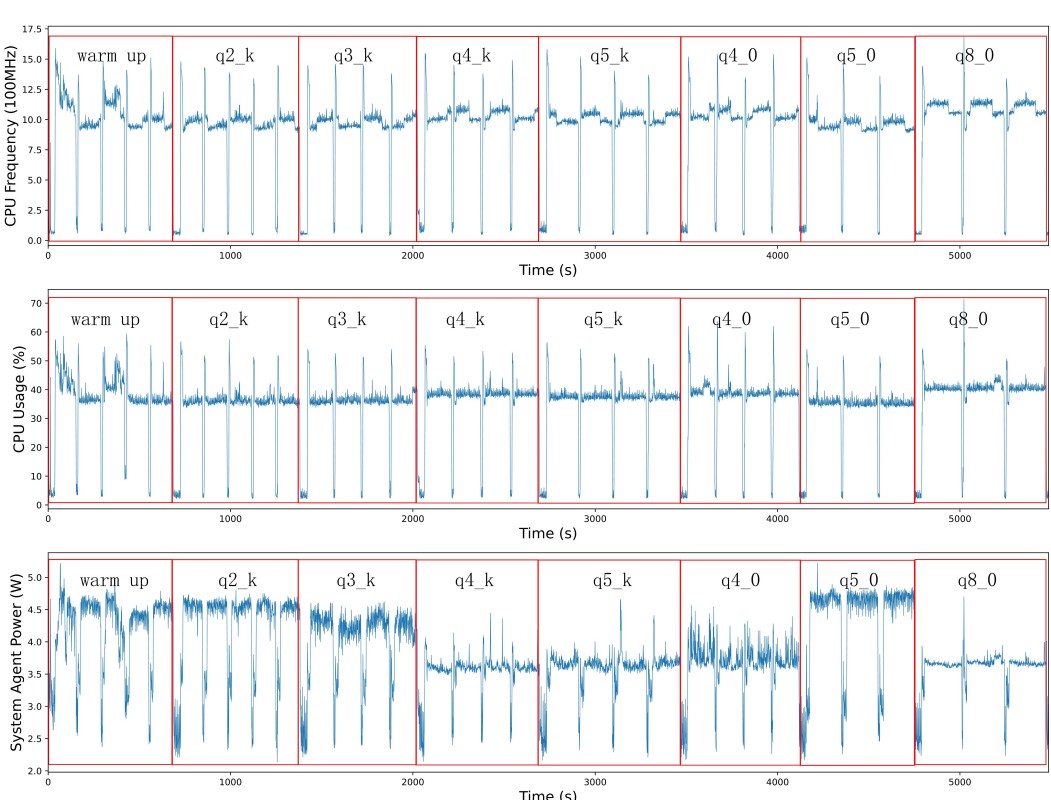

Figure 13: System Resource Utilization (CPU frequency in 100MHz, CPU utilization in percentage, System agent power in Watt) of Qwen 2.5 7B. Prompt size of 256 Tokens and Output size 1000 Tokens are fixed.

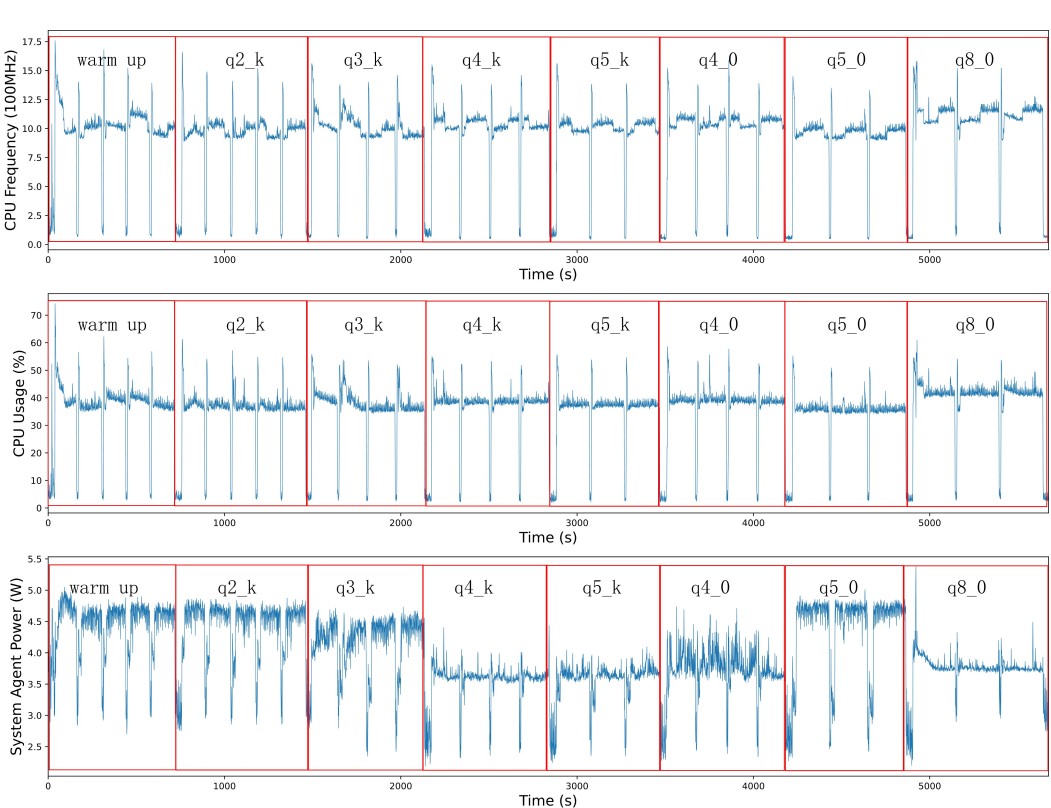

Figure 14: System Resource Utilization (CPU frequency in 100MHz, CPU utilization in percentage, System agent power in Watt) of Llama3.1 8B. Prompt size of 256 Tokens and Output size 1000 Tokens are fixed.

