# OpenReview forum: "Harnessing Large Language Models Locally: Empirical Results and Implications"
_ICLR.cc/2026/Conference — ICLR 2026 Conference Withdrawn Submission_

### Official Review · Reviewer_t4Hh · 2025-10-23

**Soundness:** 2
**Presentation:** 2
**Contribution:** 3
**Rating:** 2
**Confidence:** 5

**Summary:**

This paper presents a comprehensive empirical study of on-device Large Language Model (LLM) inference, structured around a tripartite evaluation framework covering model capability, deployment efficiency, and system resource utilization. The authors evaluate a wide range of models (0.5B to 14B from Qwen 2.5 and Llama 3) with seven post-training quantization (PTQ) methods from the llama.cpp framework on a commodity laptop. The study produces several practical insights, such as the shift from compute- to communication-bound bottlenecks for smaller models and the observation that low-level operator implementation, rather than model size, can dominate power consumption. The paper concludes by offering actionable guidance for deploying LLMs on resource-constrained devices. The paper is well-executed and practically useful, but its novelty is limited.

**Strengths:**

1. Clear and reproducible experimental setup
2. Comprehensive coverage of quantization formats available in llama.cpp, with consistent benchmarking across multiple model sizes.
Detailed engineering discussion of quantization implementation effects (e.g., unpacking costs, bit‑width overheads).
3. Practical observations that may benefit practitioners deploying LLMs to constrained CPU‑based environments.

**Weaknesses:**

1. Lack of methodological novelty: No new quantization algorithm, system architecture, or theoretical contribution is proposed; the framework is essentially a re‑organization of well‑known metrics (accuracy, throughput, resource usage) and standard benchmarks.
2. Engineering focus over scientific insight: The results largely confirm existing community knowledge (e.g., 4‑bit sweet spot, extreme compression hurts accuracy) without deeper analysis, generalization, or predictive modelling.
3. Narrow hardware scope: All edge results are on one consumer‑grade x86 laptop; no evidence that conclusions hold on ARM laptops, NPUs, mobile GPUs, or other common edge platforms.
4. ICLR fit: The paper reads as a well‑documented engineering report/benchmark rather than a research contribution with clear novelty or new understanding; this weakens its competitiveness at a method‑focused venue.

**Questions:**

1. How is the “tripartite framework” fundamentally different from standard evaluation practice in system papers?
2. Could the observed compute‑bound vs. communication‑bound regime shifts be formalized into a predictive analytical model rather than purely descriptive plots?

---

### Official Review · Reviewer_9haL · 2025-10-29

**Soundness:** 2
**Presentation:** 3
**Contribution:** 1
**Rating:** 2
**Confidence:** 4

**Summary:**

The paper aims to provide a systematic empirical evaluation framework for on-device large language models (LLMs) along three dimensions: model capability, deployment efficiency, and system resource utilization. The authors benchmark Qwen 2.5 and Llama 3 series models of varying parameter scales (0.5B–14B) with multiple post-training quantization schemes using the llama.cpp framework on commodity laptop hardware. The reported findings include scaling trends in accuracy and throughput, trade-offs between bit width and performance, and observations on compute vs. memory bottlenecks.

**Strengths:**

Comprehensive empirical benchmarking across multiple model sizes, quantization levels, and tasks.
Thorough resource utilization profiling, including CPU, memory, and power consumption.

**Weaknesses:**

The starting point of the paper is relatively basic, using existing datasets and metrics. It focuses on analyzing and summarizing experimental results, summarizing some experiences of deploying large models on the edge. These experiences already exist, and as a form of knowledge-based summary, it is feasible, but there are no particularly original or novel findings.The paper mostly analyzes results from experiments, lacking corresponding theoretical explanations, and the problems addressed do not significantly solve any challenging issues in the industry.The paper contains some errors.In section 4.2 on page 7, the last paragraph of the third line refers to Figure 2 as controlling a certain quantization method, but the paper claims that this figure gives conclusions on various precision quantization methods, which is clearly incorrect.

**Questions:**

1.Which specific pain points in the related field does the final result of this paper directly address?
2.In the last paragraph of section 4.2 on page 7, how are the relevant results of different quantization methods derived from Figure 2 in the third line?
3.Are there any novel quantization strategies or algorithmic modifications that emerged from your experiments which could be generalized?

---

### Official Review · Reviewer_ka1w · 2025-10-31

**Soundness:** 2
**Presentation:** 2
**Contribution:** 1
**Rating:** 2
**Confidence:** 4

**Summary:**

This paper conducts an empirical study on deploying large language models (LLMs) directly on consumer-grade devices using the llama.cpp framework. It evaluates multiple models (Qwen 2.5 and Llama 3, from 0.5B to 14B parameters) and seven post-training quantization (PTQ) methods across five benchmarks (GSM8K, HellaSwag, MMLU, HumanEval, and TruthfulQA).

The authors measure performance along three dimensions — model capability, deployment efficiency, and system resource utilization — using a laptop CPU setup. The study finds that system metrics roughly scale linearly with bits-per-weight, that low-bit quantized large models can outperform smaller high-precision ones, and that performance bottlenecks shift from computation to communication as models shrink. The paper concludes with deployment recommendations for balancing accuracy, speed, and resource efficiency on edge devices.

**Strengths:**

- The on-device LLM trend is rapidly gaining relevance due to privacy and cost advantages, making this study contextually important.
- Evaluates multiple models, quantization methods, and benchmarks. Uses open-source frameworks (llama.cpp, lm-evaluation-harness) and provides reproducibility details and code link.
- The tripartite evaluation framework (capability, efficiency, resource usage) provides a coherent structure. Identifies performance-power trade-offs, showing how quantization level and model scale interact in edge scenarios.

**Weaknesses:**

1. The results are confined to one hardware setup (a laptop CPU), with no exploration of whether findings generalize to heterogeneous PC CPU archs or hardware (NPU, smartphones, pad, wearable devices, etc.).
2. It only uses llama.cpp on CPUs. GPU/AI accelerator comparisons, FP8 baselines, or recent quantization methods (GPTQ, AWQ, SmoothQuant, etc.) are missing.
3. The work mainly reproduces known trends about quantization–accuracy–efficiency trade-offs without introducing new frameworks, or theoretical insights. The paper lacks engagement with prior literature on efficient LLM inference.

**Questions:**

- How were the quantization parameters (scales/zero-points) tuned? Were all methods implemented natively via llama.cpp, or were any custom modifications applied?

- Why were only CPU results reported, even though edge devices increasingly include NPUs or integrated GPUs?

- How does the chosen hardware (16GB RAM laptop) represent a general “AI PC” class—are the conclusions expected to hold for ARM-based chips or mobile SoCs?

- Did the authors compare the quantized models’ latency/energy against GPU or cloud inference baselines to contextualize results?

- Are the conclusions (e.g., 3.5 BPW threshold) robust across different datasets and architectures beyond Qwen and Llama families?

---

### Official Review · Reviewer_ck71 · 2025-10-31

**Soundness:** 2
**Presentation:** 2
**Contribution:** 2
**Rating:** 2
**Confidence:** 4

**Summary:**

This paper evaluates the deployment of large language models (LLMs) on edge devices using llama.cpp. It primarily examines two model families—LLaMA and Qwen—across different model sizes and quantization strategies. The evaluation focuses on model capability, deployment efficiency, and system resource utilization on a Windows laptop, and also analyzes the impact of varying input lengths.

**Strengths:**

* The paper explores the impact of specific implementation details in quantization strategies on model deployment.
* It presents several interesting and unexpected results, along with corresponding explanations.
* The evaluation is more comprehensive than that of similar studies in certain aspects.

**Weaknesses:**

* Most of the results are not particularly surprising.
* Some of the more interesting results cannot be reliably reproduced.
* Results obtained from a single hardware platform lack persuasiveness and limit generalizability.
* The paper does not clearly highlight its key contributions or distinguish its differences from related work.

**Questions:**

* The extra memory overhead of Q4_0 shown in Figure 3 cannot be reproduced, either using the official llama.cpp repository or the open-source code provided by the authors. Which version of llama.cpp was used in the experiments? Can the experiments—excluding those related to model capability—be reproduced on other platforms?

* Given the above issue, the generalizability of the other experimental results is questionable. Furthermore, the main distinction between this paper and similar works lies in its detailed discussion of quantization methods and the associated memory and computational bottlenecks. However, this discussion is scattered across the paper, lacking a clear structure, which makes it difficult to identify the paper's core novelty. It would be better to reorganize the paper around its main contributions, emphasizing new findings rather than reiterating existing conclusions. Nevertheless, given the limited strength of the presented data and the lack of significant differences among different comparisons, it is uncertain whether the current evidence sufficiently supports a full paper. If the authors have a stronger justification, please provide it.

---

### Note · Authors · 2025-12-28

I have read and agree with the venue's withdrawal policy on behalf of myself and my co-authors.